# “From Drowning to Treading Water”: Adolescents and Young Adults Living with Incurable and Indolent Metastatic Soft Tissue Sarcoma for More than Two Years

**DOI:** 10.3390/cancers17030442

**Published:** 2025-01-28

**Authors:** Paul R. D’Alessandro, Caitlin E. Homanick, Brittany D. Cooper, Katelyn Ferguson, Hillary Rutan, Joseph G. Pressey

**Affiliations:** 1Division of Oncology, Cancer and Blood Diseases Institute, Cincinnati Children’s Hospital Medical Center, Cincinnati, OH 45229, USA; 2College of Medicine, University of Kentucky, Lexington, KY 40506, USA; 3Department of Pediatrics, University of Cincinnati College of Medicine, Cincinnati, OH 45267, USA

**Keywords:** adolescents and young adults (AYA), oncology, psycho-oncology, sarcoma

## Abstract

This pilot study surveyed a rare cohort of adolescent/young adult (AYA) cancer patients who were living with incurable yet indolent metastatic soft tissue sarcomas. With increasing biological insight and successful use of targeted therapies, it is anticipated that this patient group will experience increasingly prolonged survival. Yet, given the rarity of the patient population, it is unclear how to best capture their experiences. Hence, we conducted a single-center pilot study using standardized symptom measures. We also asked patients to describe their illness experiences in their own words. We have included a medical summary of how they were treated. We anticipate that our research findings will be used to generate a conceptual framework that can be further studied in a larger multi-institutional study. The ultimate goal is to improve the experiences of other similar AYA patients when they interact with their health care teams.

## 1. Introduction

Approximately 89,000 adolescents and young adults aged 15–39 years (AYAs) in the United States and 900,000 AYAs worldwide are diagnosed with cancer annually [1,2]. AYAs represent a distinct population with unique care needs [1]. In particular, AYAs with metastatic soft tissue sarcoma (STS) typically face a dismal prognosis, with a median overall survival of 12–18 months [3,4]. While <20% of patients are alive at 2 years, long-term survival beyond 5 years (with or without disease present) is documented in 5–9% of patients [3,4].

Due to the rarity of diagnoses and heterogeneity within this population, a paucity of data exists regarding the illness experiences of AYAs with an indolent course. Additionally, because these patients are so few and unique, it is unclear how to best capture their experiences. These patients typically receive multiple lines of antineoplastic treatment, including precision-based therapies (i.e., targeted agents, immune checkpoint inhibitors) [3]. These therapies can have a delayed onset of action and prolonged antineoplastic effect after discontinuation, leading to clinical benefit for months to years [5,6,7,8,9,10,11,12,13,14]. Although patients are classified as long-term ‘survivors’, many are likely living with incurable metastatic disease [3]. Patient-reported outcome measures, mixed-methods research, and qualitative data have been utilized to describe the experiences of AYAs with sarcomas and other rare cancers [15,16]. However, robust measurement of the quality of life in patients with STS remains challenging due to the heterogeneity of diagnoses, subtypes, treatments, and prognoses [16,17]. Specifically, the experiences of STS patients who have a more indolent course may not be captured by existing measures. Most of these measures focus either on patients who are on active treatment for STS with acute symptoms or those who are off treatment in longitudinal surveillance programs focused on monitoring for recurrence or late effects [16,17]. A qualitative study of AYAs living with uncertain or poor cancer prognoses (UPCP) identified fluctuating needs related to physical and cognitive symptoms and psychosocial concerns over time [18]. AYAs with UPCP also described grief over their inability to achieve normative life milestones and anxiety related to future disease progression [18]. However, this study mostly included AYAs with low-grade glioma and a smaller proportion of patients with sarcoma (without identifying STS subtypes). Moreover, it did not isolate patients who had been living with UPCP for a prolonged time. There have been no published studies to date that have exclusively focused on patients living with incurable metastatic STS more than two years from diagnosis.

The primary aim of this pilot study was to characterize the experiences of patients (AYAs aged 15–39 years at diagnosis) living with incurable and indolent metastatic soft-tissue sarcoma for more than two years, utilizing patient-reported outcome measures and a qualitative open-ended question. Given the rarity and uniqueness of this patient population, the additional exploratory aim was to generate a preliminary conceptual framework regarding those illness experiences that could be tested in an expanded cohort of patients in a multi-center study.

## 2. Materials and Methods

Patients were recruited at a quaternary care children’s hospital with a comprehensive AYA oncology program. Recruitment lasted two years, from 31 January 2023 to 17 January 2025. All patients were enrolled at the onset of the study between 8 February 2023 and 12 May 2023. Patients aged 15–39 years at diagnosis who were living with incurable metastatic soft-tissue sarcoma for more than two years were eligible. Patients who were not fluent in English, and those with secondary neoplasms were excluded. Institutional outpatient clinic schedules were screened weekly, with eligible AYAs recruited during hospital visits. After obtaining informed consent, participants completed one in-person study visit that included a questionnaire and a demographic form. The questionnaire included patient-reported outcome measure information system (PROMIS) short-form questionnaires for seven domains and an open-ended question. Patient demographics and clinical variables were collected via medical record review.

### 2.1. PROMIS Measures

Patients completed quantitative PROMIS short-form questionnaires for seven domains. Domains were selected a priori following a literature review based on their strong psychometric properties and previous use in AYA sarcoma patients [19,20,21,22]. These included: Pain Interference Short Form 6a; Fatigue Short Form 7a; Physical Function Short Form 8a; Ability to Participate in Social Roles and Activities Short Form 6a; Emotional Distress Anxiety Short Form 7a; Sleep Disturbance Short Form; and Depression Short Form 6a.

### 2.2. Demographic and Clinical Information

Patient and clinical variables (age, sex assigned at birth, diagnosis, dates of diagnosis and treatments, active treatment status at time of study, and therapies received) were collected from medical records using a standardized data extraction form. Patient-reported variables (gender identity, sexual orientation, race, ethnicity, relationship status, parental status, education, employment, income, and insurance status) were obtained via a demographic form.

### 2.3. Analyses

PROMIS questionnaires were scored using standardized rubrics (US reference population normalized to T-score 50 ± 10). (https://www.healthmeasures.net/, first accessed on 1 August 2022). The proportion of patients with T-scores outside the reference range, indicating greater or less symptomatology, was reported for each domain. To supplement quantitative data, an open-ended question was asked: “Please describe your experience living with cancer. You may share any thoughts or feelings that you determine to be relevant”. Qualitative responses were transcribed and summarized by the first author (P.R.D.). Two authors (P.R.D. and C.E.H.) independently reviewed the responses to identify representative statements/quotations and generate keywords and codes. P.R.D. and C.E.H. reviewed the independently generated keywords and codes, resolved discrepancies through discussion, and generated themes that were incorporated into a preliminary conceptual framework. This process followed the inductive thematic analysis outline described by Naeem et al. 2023 [23,24].

## 3. Results

### 3.1. Patient Characteristics and Clinical Variables

Five patients were eligible, invited to participate, and completed questionnaires. Mean age was 29.4 years (18.5–39.8 years) at diagnosis and 34 years (23.2–45.7 years) at time of study. Three patients were female; two were male; four were White; and one was Black/African American. Diagnoses included low-grade *ARHGAP23::FER* spindle cell malignancy, a novel fusion-driven sarcoma [25]; *ASPSCR1::TFE3* alveolar soft part sarcoma (ASPS); INI-1 deficient epithelioid sarcoma (ES); *EWSR1::NR4A3* extra-skeletal myxoid chondrosarcoma (EMC); and *WWTR1::CAMTA1* epithelioid hemangioendothelioma (EHE).

Mean time since diagnosis was 4.5 years (2.6–6 years), and mean treatment duration was 4.2 years (range 1.5–6 years). Four patients were receiving therapy at the time of the study. On average, patients received 4.8 lines (range 2–8 lines) of antineoplastic therapy over 4.2 years (range 1.5–6 years). All patients received at least one targeted therapy or immune checkpoint inhibitor, monotherapy, or in combination. Four patients received multiple kinase or tyrosine kinase inhibitors, including pazopanib (n = 4); sunitinib (n = 2); axitinib (n = 2); and cabozantanib (n = 1). Two patients received conventional cytotoxic chemotherapy. The patient with low-grade *ARHGAP23::FER* spindle cell malignancy initially received two cycles of methotrexate and vinorelbine and was switched to lorlatinib (an ALK/ROS/FER inhibitor) once the actionable fusion was identified. The patient with INI-1-deficient epithelioid sarcoma received palliative liposomal doxorubicin as an eighth line of therapy after multiple unacceptable toxicities and disease progression events. Three patients received radiation therapy. Notably, the patient with EHE was enrolled in a clinical trial and received FLASH proton therapy for upper extremity bone pain [26]. In addition to targeted therapy, the patient with extra-skeletal myxoid chondrosarcoma underwent palliative below-knee amputation for burdensome disease as well as pulmonary metastatectomy (Table 1, Figure 1).

### 3.2. T-Scores for PROMIS Measures

The T-scores for all patients and for all measures are summarized in Table 2. Patient 1, with *ARHGAP23::FER* malignancy, reported above-average functioning in terms of fatigue, physical functioning, and participation in social roles and activities. Patient 2, with ASPS, reported no symptoms outside of the U.S. adult reference normative ranges. Patient 3, with ES, reported increased pain and fatigue, decreased physical function, and greater sleep interference. Patient 4, with EMC, reported increased fatigue, anxiety, and depression, and decreased physical function and participation in social roles and activities. Patient 5, with EHE, reported increased anxiety.

### 3.3. Qualitative Responses

Two authors (P.R.D. and C.E.H.) independently reviewed the quotations to generate keywords and codes. P.R.D. generated 7 codes, and C.E.H. generated 8 codes. Agreement/overlap was reached for 12 of 15 codes (80%). Identified discrepancies (3/15 or 20% of codes) were resolved with discussion and consultation with the original data source. From these codes, four overarching themes were identified by both authors: managing physical symptoms related to disease and/or treatment side effects; navigating feelings of guilt and inadequacy; changing illness experience over time; and self-reflection generating gratitude. Themes, codes, and representative statements/quotations are summarized in Table 3. A concept map was then generated that included themes, codes, and keywords. As a final exploratory exercise, PROMIS measures that overlapped with the themes generated from the coding analysis were added into the concept map at the level of codes or keywords. For instance, pain, fatigue, and decreased physical function overlapped with “managing physical symptoms related to disease or treatment side-effects”. Additionally, anxiety and participation in social roles and activities overlapped with “navigating feelings of guilt and inadequacy” under the grief/worry and social comparison codes, respectively (Figure 2).

## 4. Discussion

The present study described a cohort of AYA patients living with indolent metastatic soft tissue sarcomas (STS) for at least two years. STS are a group of rare and heterogeneous tumors with more than 70 different histological subtypes, accounting for approximately 8% of AYA cancers [17,27]. Although prognosis for patients with metastatic STS is dismal, a subset of AYAs is alive beyond two years from diagnosis [4]. Despite a two-year recruitment period at a high volume center (approximately 100 new AYA diagnosis per year), only five patients were eligible. All were recruited at the start of the study period. No additional patients reached eligibility criteria of living for two years from diagnosis despite the two-year recruitment period. These data speak to the rarity and uniqueness of this subset of patients. In order to generate more robust and meaningful data, a larger multi-center study will be required in order to recruit more patients. We present our preliminary data at this juncture with an iterative aim to generate concepts and hypotheses that could be further explored in a future collaborative study.

Each AYA patient in our cohort had a different histologic subtype of STS. In larger series, characteristics associated with long-term survival from diagnosis of metastatic STS included female gender; primary tumors < 5 cm; simple genetic alterations; and certain histologic subtypes, including EHE and low-grade tumors [3,4]. Some patients in our cohort exhibited better prognostic features, including three females; a patient with EHE; and a patient with low-grade spindle cell malignancy with a novel targetable fusion.

Our patients’ treatments, with multiple lines of therapy over many years, were reflective of previously published literature. In a series of patients with STS who were alive five years after the diagnoses of metastases, 46% of patients were treated with five or more lines of treatment, and 20% of patients received eight or more lines of treatment, typically involving clinical trial enrollment [4]. Clinical trial enrollment has been associated with prolonged progression-free survival [12].

Molecular profiling of all but one tumor in our cohort was utilized to confirm the diagnosis and tailor therapeutic decisions. Consensus guidelines advise clinicians to test tumor tissue for specific genetic alterations and molecular fusions upfront or at relapse in STS patients with surgically unresectable or metastatic disease to identify targetable alteration(s) [8]. Such therapies can have a delayed onset of action and/or prolonged antineoplastic effect after discontinuation, leading to clinical benefit for months to years [5,6,7,8,9,10,11,12,13,14]. Some agents have been utilized as chronic maintenance therapy [5].

One patient in our cohort had treatment interrupted to facilitate a spontaneous pregnancy with healthy live birth at term (Figure 1). Despite persistent metastatic disease, she has now successfully carried a subsequent pregnancy to term. Women with metastatic cancer rarely consider pregnancy or parenthood because of uncertain life expectancy, fear of cancer progression, incompatible treatments, or uncertainty regarding the effects of cancer treatments on the fetus [4]. A case series described four women with metastatic, unresectable STS who started and carried first pregnancies while living with indolent disease [4]. Each patient was at least four years from diagnosis with good functional status. Two pregnancies were facilitated with in vitro fertilization. All women stopped systemic cancer treatment during pregnancy. Three women had documented disease progression during treatment interruption, but all had tumor control re-established after reintroduction of chemotherapy following pregnancy. All four patients were long-term survivors (20 months to two decades) [4]. These data highlight the possibility of treatment interruptions to facilitate pregnancy in highly selected subsets of female patients with metastatic STS.

PROMIS measures allowed comparison of each patient to a reference population to gauge symptom burden. Patients reported increased pain, fatigue, sleep interference, and anxiety, with decreased physical function and social participation. Pain, fatigue, and decreased physical function are commonly reported in patients with STS, regardless of diagnosis or treatment modality/duration, due to tumor location (i.e., often localized to an extremity) [17]. Anxiety is also a symptom that has been well described by other AYAs with UPCP. This anxiety is multifactorial, begotten by isolation, loneliness, and fear [18]. Patients describe fear—for self and for loved ones—related to future disease progression [18]. Of note, Patient 1 (with *ARHGAP23::FER* malignancy) achieved disease control with an oral targeted agent. This patient reported above-average function compared to the adult U.S. reference population in three measures (pain, physical function, and social participation, Table 2). As described above, such therapies may represent novel treatment strategies for patients with STS and have been associated with long-term clinical benefit, particularly when used as chronic maintenance therapy [5]. Although all measures were endorsed by at least one patient, no measure was endorsed by more than two patients within the cohort. Though this may represent heterogeneity within the cohort, it is also possible that PROMIS measures were inadequate to capture the patients’ illness experiences.

Responses to the open-ended question overlapped with some PROMIS domains but also revealed concepts that were not captured in the quantitative measures (Table 3). Common themes included: managing physical symptoms related to disease and/or treatment side effects; navigating feelings of guilt and inadequacy; changing illness experience over time; and self-reflection generating gratitude. Physical symptoms and treatment-related side effects included pain, strength/mobility, and fatigue. Additional cognitive symptoms related to medications were also reported. Patients’ guilt and inadequacy not only related to their physical symptoms but were contextualized within their perceived self-image as AYAs. Some patients were parents/caregivers themselves and felt guilty that their own physical limitations prevented them from fully caring for their children (i.e., “I felt like a bad mom”). Other patients, diagnosed at a time when they sought more independence, felt “shocked” that they had to rely on their parents or family members for physical support. This inadequacy also extended to social comparisons, including comparisons to their pre-illness selves, other patients living with cancer, and other people who were not living with cancer (via social media). Patients described fluctuating illness experiences over time. One patient reported that they “never thought [they] would live so long”, at times, “forget [ting] that they had cancer”. Another patient likened their illness experience to “drowning” at the time of diagnosis and “treading water… to… stay afloat” several years later. Changing needs related to physical symptoms (i.e., disease or treatment-related side effects) and psychosocial concerns (i.e., persistent worry about disease progression or ongoing mortality confrontation) have been described in AYAs with STS and/or UPCP [18]. However, our cohort of patients described distinct mindset shifts after two years, characterized by adjustment and acceptance. This finding is particularly novel, as no studies to date have specifically described AYAs living with incurable, metastatic STS for such a long duration.

Lastly, patients reported that their long illness duration gave them perspective and time to reflect. These reflections manifested as both internal self-discoveries as well as external expressions of gratitude for opportunities that they did not expect to have (including personal and vocational goals). Although some AYAs with UPCP experience grief over their inability to achieve normative life milestones, others describe feelings of urgency to prioritize enjoyable activities (i.e., travel) or life goals [18]. Patients who reframe their situation to maximize “time they have left” often feel gratitude for that time [18]. For our patients, the “time they [had] left” was longer than they expected, which may have been a driving factor for these perceptions.

Our patients also extended gratitude to their multidisciplinary health care team, suggesting that they perceived benefit to receiving comprehensive care at a center with sarcoma and AYA expertise. One patient reflected: “Cancer has shown me how far past my limits I can push myself… Real pain, real love, real discipline, real courage. Not just in me, but in others… especially the children I have met [at the children’s hospital]. And my doctors. Champions all”. This patient cited their specific environment (surrounded by much younger pediatric cancer patients who were going through similar experiences) as particularly motivating and inspiring. Embedding AYA care within pediatric hospitals (i.e., treating AYAs with pediatric protocols at pediatric loci of care compared to pediatric protocols at adult loci of care) has been associated with improved survival outcomes for AYAs with leukemia at Canadian centers [28]. It is suggested that superior outcomes may relate to expanded supportive care that AYAs receive within a comprehensive pediatric/AYA program [28]. It is possible that comprehensive supportive care designed to meet AYA needs at such unique loci of care also influences patient-reported illness experiences.

Although full qualitative analysis is beyond the scope of our preliminary study, these phenomena could be elucidated in further work that is expanded to other sites. Given the anticipated small sample size at individual sites, it is likely that several sites of comparable size would be needed for collaboration. The most feasible methods may be purely qualitative (i.e., interviews) conducted centrally via a study team at a single institution. In this instance, virtual participation may be needed.

### Limitations

This study was limited by a small cohort of patients with inherently rare, heterogeneous STS. Two patients died after participation, limiting the ability to revise informed consent for study measures or conduct follow-up research with this cohort. Results may not be generalizable to larger AYA populations or other patients with STS. This study was not adequately powered to assess for differences in patient-reported outcomes based on clinical variables. Finally, this study was not designed to assess patients across multiple time points. Additional work is needed to understand the trajectory of symptoms and concerns in this population over time.

## 5. Conclusions

Despite a two-year recruitment period at a quaternary center, only five AYA patients living with incurable metastatic soft tissue sarcoma for more than two years were identified. Patients were treated with multiple lines of antineoplastic therapy longitudinally, including targeted agents and immune checkpoint inhibitors. Via PROMIS measures, patients reported increased pain, fatigue, and anxiety, with decreased physical function and ability to participate in social roles and activities. Exploratory thematic analysis of qualitative data identified additional common illness experiences, including feelings of guilt and inadequacy; distinct shifts in experiences over time; and self-reflection generating gratitude. These data can serve as the foundations for a larger, multi-center study with the aim of capturing the experiences and needs of this rare and unique subset of patients.

## Figures and Tables

**Figure 1 cancers-17-00442-f001:**
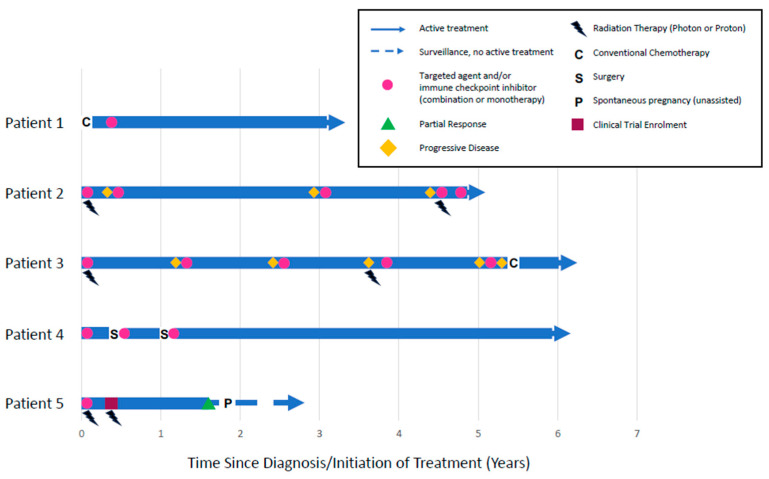
Swimmer plot describing the treatments for five patients living with incurable metastatic soft tissue sarcoma for more than two years. Diagnoses included low-grade *ARHGAP23::FER* spindle cell malignancy (Patient 1); *ASPSCR1::TFE3* alveolar soft part sarcoma (Patient 2); INI-1 deficient epithelioid sarcoma (Patient 3); *EWSR1::NR4A3* extra-skeletal myxoid chondrosarcoma (Patient 4); and *WWTR1::CAMTA1* epithelioid hemangioendothelioma (Patient 5). Mean time since diagnosis was 4.5 years (range 2.6–6 years), and mean treatment duration was 4.2 years (range 1.5–6 years). On average, patients received 4.8 lines (range 2–8 lines) of antineoplastic therapy over 4.2 years (range 1.5–6 years). All patients received at least one targeted therapy or immune checkpoint inhibitor.

**Figure 2 cancers-17-00442-f002:**
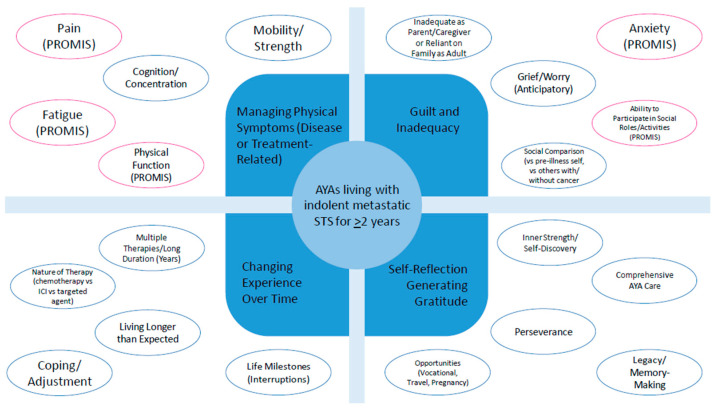
Concept map. An exploratory thematic analysis was conducted in order to generate themes that could be studied in a multi-center expansion cohort with a larger sample of patients. Four main themes were identified, with surrounding illustrative codes and keywords. PROMIS measures that overlapped with the codes or keywords were also inserted into the conceptual framework.

**Table 1 cancers-17-00442-t001:** Patient characteristics and clinical variables.

Patient Characteristics	n (%)
Age at diagnosis, years, mean (range)	29.5 (18.5–39.8)
Age at time of survey, years, mean (range)	34.0 (23.2–45.7)
Sex	
Male	2 (40)
Female	3 (60)
Race	
White	4 (80)
Black	1 (20)
Ethnicity	
Non-Hispanic	5 (100)
Sexual Orientation ^a^	
Heterosexual	4 (100)
Relationship Status ^a^	
Single	1 (25)
In a Relationship	3 (75)
Parental Status	
More than one child	3 (60)
No children	2 (40)
Highest Level of Education Completed	
High School	2 (40)
College/Vocational	1 (20)
Graduate/Doctoral Degree	2 (40)
Education Status, current	
Full-Time Student	1 (20)
Part-Time Student	1 (20)
Not Currently in School	3 (60)
Employment Status, current	
Full-Time Work (≥40 h per week)	1 (20)
Part-Time Work (<40 h per week)	1 (20)
Employed, but Not Working	1 (20)
Not Currently Employed	2 (40)
Self-Reported Income Quintile	
1st Quintile (<$27,000 USD)	1 (20)
2nd Quintile ($27,000–$51,999 USD)	1 (20)
3rd Quintile ($52,000–$84,999 USD)	1 (20)
4th Quintile ($85,000–$140,000 USD)	2 (40)
5th Quintile (>$140,000 USD)	0 (0)
Insurance Status	
Private	4 (80)
Public, including Medicaid	1 (20)
Uninsured	0 (0)
**Treatment Characteristics**	**n (%)**
Time since diagnosis, years, mean (range)	4.5 (2.6–6)
Active treatment duration, years, mean (range)	4.2 (1.5–6)
On active treatment	4 (80)
Incisional or excisional biopsy	5 (100)
Number of lines of therapy, mean (range) ^b^	4.8 (2–8)
Conventional chemotherapy	2 (40)
Surgery	1 (20)
Radiation therapy, photon or proton	3 (60)
Targeted therapy and/or immune checkpoint inhibitor	5 (100)
Enrollment in clinical trial	1 (20)

^a^ n = 1 missing; ^b^ including conventional chemotherapy, surgery, radiation therapy, targeted therapy, and immune checkpoint inhibitors, excluding biopsy.

**Table 2 cancers-17-00442-t002:** T-scores for PROMIS Measures. PROMIS measure T-score results for each patient and each domain are summarized. Questionnaires were scored using standardized rubrics (US reference population normalized to T-score 50 ± 10, https://www.healthmeasures.net/), first accessed on 1 August 2022). T-scores > 60 were considered having greater or worse symptomatology compared to the general US adult population, except for physical function and ability to participate in social roles and activities, which were scored in reverse (indicated in red). T-scores < 40 were considered as having lesser or improved symptomatology compared to the general US adult population, except for physical function and ability to participate in social roles and activities, which were scored in reverse (indicated in green).

	Pain T-Score	Fatigue T-Score	Physical Function T-Score	Social Participation T-Score	Anxiety T-Score	Sleep T-Score	Depression T-Score
Patient 1	52	**36.9**	**61.3**	**65**	52.6	41.7	38.4
Patient 2	58.2	57.8	41.9	51.9	49.9	50.4	59.3
Patient 3	**66.7**	**64.8**	**38.9**	41.6	55.1	**66.4**	48.3
Patient 4	50.7	**67.8**	**35.5**	**37.2**	**63.8**	56.8	**60.5**
Patient 5	54.5	55.1	43	45.6	**66.4**	50.4	53.2

**Table 3 cancers-17-00442-t003:** Preliminary thematic analysis of responses to open-ended question, including codes and illustrative quotations/statements.

Theme	Code	Illustrative Quotations/Statements
Managing physical symptoms related to disease/treatment side effects	Disease-related symptoms	“I was constantly tired, in pain. I am upset because my mobility and my strength in my arm [are] very bad. [I experience] exhaustion…” “[Around the time of diagnosis], I lost my ability to walk…adjusting to my reduced mobility”.
	Treatment-related side effects	“My main problem is the fatigue. My medicine affects how I feel, but I am usually tired. It clouds my thoughts and my words. A lot of times, I can’t remember what I want to say mid-sentence…”
Navigating feelings of guilt and inadequacy	Guilt	“I felt very down… I can’t always get down on the floor and play with my children…I sometimes feel like I’m a bad mom”. “I was shocked that I couldn’t handle [my usual] tasks and responsibilities… my family was a great help in picking up the slack”.
	Grief/worry (anticipatory)	“I do fear [for my family members] when we lose control of the disease…It’s really tough to remember all the plans we had for our future”. “I worry about being able to play [with my children] when they get older”.
	Social comparison	“I miss working and being social at the same level as before. I try not to have a ‘victim’ mindset, but I do compare myself up against “normal” people without cancer or [to] others’ cancer journeys… Several people I’ve seen on social media are like “woe is me”.
Changing illness experience over time	Long treatment duration	“It’s really still kind of crazy, even after almost 6 years. There are times when I forget that I have cancer…I never thought I’d live so long and feel ok”.
	Changes to experience	“Personal experience can change over time. It was overwhelming at first; [and this feeling] continued for another year or two… After about 2.5–3 years of treatment, things stopped being overwhelming. Instead of feeling like I was … drowning, I [began to feel like I was] treading water and trying to stay afloat”.
Self-reflection generating gratitude	Perseverance (inner strength)	“Cancer has shown me how far past my limits I can push myself… Real pain, real love, real discipline, real courage. Not just in me, but in others… especially the children I have met [at the children’s hospital]. And my doctors. Champions all”.
	Comprehensive AYA care (locus of care)	“I am grateful for my team here. I’m not sure I would be doing as well if I were somewhere else”.
	Opportunities	“I also wanted more kids and didn’t know if that was going to be possible”. I try to take advantage of the good days… We… have been travelling. Recovery from travelling is tough, but the memories are worth it”.

## Data Availability

The datasets used and/or analyzed during the current study are available from the corresponding author by request.

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
