# Peer review of "“From Drowning to Treading Water”: Adolescents and Young Adults Living with Incurable and Indolent Metastatic Soft Tissue Sarcoma for More than Two Years"

_cancers, 2025, doi:10.3390/cancers17030442_

Round 1
Reviewer 1 Report
Comments and Suggestions for Authors
This manuscript addresses adolescent and young adult (AYA) patients' experiences with low-grade metastatic soft tissue sarcoma (STS), a significant yet understudied topic in oncology. While the inclusion of both quantitative QOL metrics and qualitative responses is commendable and presents an opportunity to contribute meaningfully to the literature on this rare patient population, the manuscript in its current form has several substantial limitations that preclude its publication in this journal.
Key Concerns
1.Sample Size and Generalizability
The study is significantly constrained by its small sample size of only five participants. While we acknowledge the rarity of this patient population, this limitation substantially restricts the robustness of quantitative analyses, reduces statistical power, and severely compromises the ability to generalize findings. It is essential that the manuscript be appropriately framed as an exploratory or hypothesis-generating study, particularly given these data limitations.
2.Insufficient Depth of Qualitative Analysis
While the inclusion of qualitative responses adds value to the manuscript, their analysis lacks depth and systematic methodological rigor. Although themes are identified, they are neither adequately examined nor contextualized within the broader psychosocial oncology literature, diminishing the interpretability and impact of the findings.
3. Inadequate Discussion and Implications
The discussion section fails to properly integrate the findings with existing knowledge and inadequately addresses implications for clinical practice. Furthermore, it does not present a compelling argument for how this research advances the field of oncology or improves care for AYA patients with STS.
4.Limited Results
Both the quantitative findings and qualitative insights are insufficiently detailed, failing to provide a compelling narrative. Consequently, the manuscript does not meet the journal's standards for rigor and comprehensiveness.
Recommendations
While the study addresses an important topic, its methodological, sample size, and analytical limitations are substantial. Under these circumstances, acceptance of the manuscript is not possible. We strongly recommend that the authors undertake major revisions, including:
- Expanding the study cohort
- Deepening the qualitative analysis and improving methodological rigor
- Revising the discussion to emphasize clinical relevance and innovation
Through these improvements, we hope this research can enhance understanding of this rare AYA patient population and advance knowledge in the field of oncology.
Author Response
This manuscript addresses adolescent and young adult (AYA) patients' experiences with low-grade metastatic soft tissue sarcoma (STS), a significant yet understudied topic in oncology. While the inclusion of both quantitative QOL metrics and qualitative responses is commendable and presents an opportunity to contribute meaningfully to the literature on this rare patient population, the manuscript in its current form has several substantial limitations that preclude its publication in this journal.
Key Concerns:
- Sample Size and Generalizability: The study is significantly constrained by its small sample size of only five participants. While we acknowledge the rarity of this patient population, this limitation substantially restricts the robustness of quantitative analyses, reduces statistical power, and severely compromises the ability to generalize findings. It is essential that the manuscript be appropriately framed as an exploratory or hypothesis-generating study, particularly given these data limitations.
- Insufficient Depth of Qualitative Analysis: While the inclusion of qualitative responses adds value to the manuscript, their analysis lacks depth and systematic methodological rigor. Although themes are identified, they are neither adequately examined nor contextualized within the broader psychosocial oncology literature, diminishing the interpretability and impact of the findings.
- Inadequate Discussion and Implications: The discussion section fails to properly integrate the findings with existing knowledge and inadequately addresses implications for clinical practice. Furthermore, it does not present a compelling argument for how this research advances the field of oncology or improves care for AYA patients with STS.
- Limited Results: Both the quantitative findings and qualitative insights are insufficiently detailed, failing to provide a compelling narrative. Consequently, the manuscript does not meet the journal's standards for rigor and comprehensiveness.
Recommendations
While the study addresses an important topic, its methodological, sample size, and analytical limitations are substantial. Under these circumstances, acceptance of the manuscript is not possible. We strongly recommend that the authors undertake major revisions, including:
- Expanding the study cohort
- Deepening the qualitative analysis and improving methodological rigor
- Revising the discussion to emphasize clinical relevance and innovation
Through these improvements, we hope this research can enhance understanding of this rare AYA patient population and advance knowledge in the field of oncology.
Response: Thank you for your thoughtful comments and the opportunity to improve our manuscript. Regarding the sample size and request to expand the cohort, our quaternary center runs a comprehensive, high volume AYA oncology program (approximately 100 new AYA cancer diagnoses per year.) Our study has remained open to recruitment for 2 years, but only 5 patients have been eligible (all of whom were recruited at the time the study opened). No additional patients have met the inclusion criteria. We have clarified our Methods and Results sections in order to explicitly state this. In order to expand the cohort, a mutli-site collaborative study would need to be undertaken, involving several large cancer centers. Our aim in publishing our data at this juncture is to help facilitate such a collaboration. We hope that our preliminary results can be used to identify hypotheses for future work, and feasible methods by which to test them. Ultimately, as Reviewer 3 noted, this is an ultra-rare and unique subset of AYA STS patients whose experiences and voices could be diminished or lost if they are lumped in with other AYA cancer populations or other populations of patients with STS in order to mitigate a sample size.
The Methods and Results now include an inductive thematic analysis, with two independent authors, coding qualitative responses to generate keywords, codes, and themes. PROMIS data is now represented separately in Table 2, with Table 3 dedicated exclusively to qualitative data, themes, and codes. We have created a new concept map that deepens the qualitative analysis, and connects this data with overlapping PROMIS measures. The Discussion has been revised and expanded to connect with these additional analyses, and has also been expanded to emphasize clinically relevant and novel concepts (including distinct shifts in illness experiences 2 years from diagnosis and patient perceptions of treatment at a comprehensive AYA center.)
Reviewer 2 Report
Comments and Suggestions for Authors
This is a small mixed methods study of adolescent/young adults (AYA) who have been living with non-curable soft tissue sarcomas which are a rare cancer. The study entail 5 participants who complete surveys and an interview and medical data were obtained by medical record review. Because this is a rare cancer, the sample size is small from a comprehensive AYA oncology program.
The abstract is complete with some background, the aim of this pilot study, the inclusion criteria, methods and demographic results including the diagnosis and treatment. There is a fairly detailed information on the treatment they received. There seems to be less emphasis on the results or the PROMIS questionnaires. Qualitative responses appears to be summarized adequately.
The introduction provides adequate information on the prevalence of sarcoma in the AYA population. The authors note that the information from this sample may be helpful for developing novel cancer therapies, but it is not clear how this information will be helpful. Can a bit more be said about that? The aim of the study is clearly stated to characterize the experiences of patients living with incurable and indolent metastatic soft-tissue sarcoma. While it is mentioned in the abstract that this is a pilot study, it is not noted that this is a pilot study in the aim or introduction.
The methods and materials are clearly described. The inclusion criteria were noted, and the survey instruments and qualitative component and medical review are described well.
The results of the patient characteristics and clinical variables are described in detail including treatment that each received. The description of the survey results only reported the number of patients who reported general symptoms compared to the reference population. It would be helpful to remind the readers which of the surveys the results were from. A more detailed report of the results of the surveys would be of interest to readers. There is some information of the quotes to support the PROMIS domains and some results are presented there and seem more comprehensive than those on page 5 lines 160-161.
The qualitative results are a bit confusing. It appears that quotes were chosen to support the PROMIS domains. Is that the way the coding was achieved? Table 2 is a bit confusing. It appears that these were only two themes identified that were not part of the PROMIS survey packet. This would benefit from better explanation or perhaps a different display of the qualitative data.
If this is a pilot project, there should be an analysis if this type of study is feasible, etc. That could be included if indeed this is a pilot. The question to answer then is how do you know if you should move forward to a larger study.
Limitations are clearly stated. This is a heterogeneous and small sample, but with reasonable explanation.
The conclusions seem to focus on the treatments they received much more than what it is like for the patients. There is only one sentence related to the aim of the study. Can there be a bit more related to the findings related what the patients reported?
Author Response
Comment: This is a small mixed methods study of adolescent/young adults (AYA) who have been living with non-curable soft tissue sarcomas which are a rare cancer. The study entail 5 participants who complete surveys and an interview and medical data were obtained by medical record review. Because this is a rare cancer, the sample size is small from a comprehensive AYA oncology program. The abstract is complete with some background, the aim of this pilot study, the inclusion criteria, methods and demographic results including the diagnosis and treatment. There is a fairly detailed information on the treatment they received. There seems to be less emphasis on the results or the PROMIS questionnaires. Qualitative responses appear to be summarized adequately.
Response: The results from the PROMIS questionnaires are now emphasized in the Abstract.
Comment: The introduction provides adequate information on the prevalence of sarcoma in the AYA population. The authors note that the information from this sample may be helpful for developing novel cancer therapies, but it is not clear how this information will be helpful. Can a bit more be said about that?
Response: Given that a description of the impact novel therapeutics on the illness experiences of AYAs with STS is not an immediate goal of this study, this comment is now removed from the Introduction.
Comment: The aim of the study is clearly stated to characterize the experiences of patients living with incurable and indolent metastatic soft-tissue sarcoma. While it is mentioned in the abstract that this is a pilot study, it is not noted that this is a pilot study in the aim or introduction.
Response: The Introduction now accurately situates this study as a pilot study, with justification for the presentation of the data at this juncture.
Comment: The methods and materials are clearly described. The inclusion criteria were noted, and the survey instruments and qualitative component and medical review are described well.
Response: Thank you for this comment.
Comment: The results of the patient characteristics and clinical variables are described in detail including treatment that each received. The description of the survey results only reported the number of patients who reported general symptoms compared to the reference population. It would be helpful to remind the readers which of the surveys the results were from. A more detailed report of the results of the surveys would be of interest to readers. There is some information of the quotes to support the PROMIS domains and some results are presented there and seem more comprehensive than those on page 5 lines 160-161.
Response: The Results for each patient and each PROMIS measure are now summarized in the revised Table 2, which provides greater detail.
Comment: The qualitative results are a bit confusing. It appears that quotes were chosen to support the PROMIS domains. Is that the way the coding was achieved? Table 2 is a bit confusing. It appears that these were only two themes identified that were not part of the PROMIS survey packet. This would benefit from better explanation or perhaps a different display of the qualitative data.
Response: The PROMIS measures are now presented separately (Table 2.) The Methods and Results sections now include a more in-depth analysis and detailed representation of the qualitative data (Table 3.) An additional concept map, with conceptual framework is also included, highlighting which PROMIS measures overlap with codes and keywords generated from the qualitative responses to the open-ended question (Figure 2.)
Comment: If this is a pilot project, there should be an analysis if this type of study is feasible, etc. That could be included if indeed this is a pilot. The question to answer then is how do you know if you should move forward to a larger study. Limitations are clearly stated. This is a heterogeneous and small sample, but with reasonable explanation.
Response: The Discussion now includes, just prior to Limitations, an outline of potential study design components for a larger expansion cohort conducting at multiple centers.
Comment: The conclusions seem to focus on the treatments they received much more than what it is like for the patients. There is only one sentence related to the aim of the study. Can there be a bit more related to the findings related what the patients reported?
Response: Conclusions now cover both clinical characteristics and illness experiences, with greater emphasis on PROMIS measures and themes generated from qualitative data.
Reviewer 3 Report
Comments and Suggestions for Authors
This manuscript describes experiences of five AYA-patients with incurable and indolent metastatic soft tissue sarcomas, all of whom were at least two years from initial diagnosis.
All patients had received multiple therapies (average 4.8 lines of treatment), and all were still (or again) under treatment at the time of the study.
Both quantitative PROMIS questionnaires for the (pre-defined) seven domains of physical function, emotional distress/anxiety, depression, fatigue, sleep disturbances, pain interference, and ability to participate in social roles and activities, and qualitative interviews were reported in the manuscript.
The authors report that two (of five) patients, respectively, described increased fatigue, anxiety, and decreased physical function in comparison to the reference population. Moreover, in qualitative interviews, patients described uncertainty regarding the future, changing symptom burdens and illness experience over time, but also gratitude for the time they had and for goals achieved while under treatment, and towards their treatment teams. Moreover, the number and duration of treatments applied to patients are impressive. Also of interest is the observation that in these few patients, there seemed to be a shift of perception after two years from diagnosis, as reflected in the manuscript title, “From drowning to threading water”.
The manuscript is well written, easy to read and to understand, and clearly structured.
Methods and results are clearly presented. Results are adequately discussed, including the limits, and the conclusions drawn seem justified.
Even with only this small number of patients included, I find this manuscript important. This is a rare and special subset of patients, both in terms of the rare diagnosis of living with a soft tissue sarcoma with uncertain or poor cancer prognosis, and with regard to the specific AYA age group. There is much to learn about how these patients feel and what they need from this manuscript.
I would like to encourage the authors only to consider some minor thoughts:
- With only five subjects studied, I would have preferred to see more “raw data” on each patient, rather than or in addition to descriptional statistics (e.g., each single patient’s age given rather than median and ranges).
- In their discussion, the authors might make some statement about what they think how the specific setting of their institution (dedicated AYA cancer service at a quarternary children’s hospital) might have had an impact on their patients’ outcomes and attitudes. This setting is rather specific, and unfortunately rather uncommon as well. Maybe it is worth discussing how this setting might be desirable for such patients?
- Looking at the patients’ insurance status (four of five with private insurance), and at the number of treatments given, there is one slightly unpleasant thought far in the back of my head, about an eventual selection bias. Maybe the authors could have a look at the insurance status of patients being diagnosed with comparable disease status but not included in this study, because they were not alive anymore at two years after initial diagnosis?
Author Response
This manuscript describes experiences of five AYA-patients with incurable and indolent metastatic soft tissue sarcomas, all of whom were at least two years from initial diagnosis. All patients had received multiple therapies (average 4.8 lines of treatment), and all were still (or again) under treatment at the time of the study. Both quantitative PROMIS questionnaires for the (pre-defined) seven domains of physical function, emotional distress/anxiety, depression, fatigue, sleep disturbances, pain interference, and ability to participate in social roles and activities, and qualitative interviews were reported in the manuscript.The authors report that two (of five) patients, respectively, described increased fatigue, anxiety, and decreased physical function in comparison to the reference population. Moreover, in qualitative interviews, patients described uncertainty regarding the future, changing symptom burdens and illness experience over time, but also gratitude for the time they had and for goals achieved while under treatment, and towards their treatment teams. Moreover, the number and duration of treatments applied to patients are impressive. Also of interest is the observation that in these few patients, there seemed to be a shift of perception after two years from diagnosis, as reflected in the manuscript title, “From drowning to threading water”. The manuscript is well written, easy to read and to understand, and clearly structured. Methods and results are clearly presented. Results are adequately discussed, including the limits, and the conclusions drawn seem justified. Even with only this small number of patients included, I find this manuscript important. This is a rare and special subset of patients, both in terms of the rare diagnosis of living with a soft tissue sarcoma with uncertain or poor cancer prognosis, and with regard to the specific AYA age group. There is much to learn about how these patients feel and what they need from this manuscript.
Response: Thank you. We agree that this is a rare and special subset of patients, and that there is much to learn about how these patients feel and what they need.
Comment: With only five subjects studied, I would have preferred to see more “raw data” on each patient, rather than or in addition to descriptional statistics (e.g., each single patient’s age given rather than median and ranges).
Response: As per the procedures reviewed and approved by our institutional research ethics board, we were required to present demographic data in aggregate form to limit potential patient identification, recognizing that these risks are higher in a cohort of patients with ultra-rare tumors. Additionally, because two patients died since participating in the study, we cannot revise the study procedures posthumously. Having said this, in addition to the descriptive statistics, we have presented each patient’s clinical course individually via swimmer plot (Figure 1.) We have also re-formatted our Results to present each patient’s PROMIS measures individually, which allows for a more granular representation of individual patient ‘raw data’ (revised Table 2.)
Comment: In their discussion, the authors might make some statement about what they think how the specific setting of their institution (dedicated AYA cancer service at a quarternary children’s hospital) might have had an impact on their patients’ outcomes and attitudes. This setting is rather specific, and unfortunately rather uncommon as well. Maybe it is worth discussing how this setting might be desirable for such patients?
Response: Thank you. We now highlight this in the Discussion, referencing one patient’s quotation regarding being inspired and motivated by much younger pediatric cancer patients they encountered while undergoing treatment. We also reference studies that have demonstrated improved outcomes for AYA patients treated on pediatric protocols at pediatric loci of care (compared to pediatric protocols given at adult loci of care.) It is likely that the additional pediatric/AYA supportive care that AYAs receive within a comprehensive AYA program influence their attitudes and outcomes.
Comment: Looking at the patients’ insurance status (four of five with private insurance), and at the number of treatments given, there is one slightly unpleasant thought far in the back of my head, about an eventual selection bias. Maybe the authors could have a look at the insurance status of patients being diagnosed with comparable disease status but not included in this study, because they were not alive anymore at two years after initial diagnosis?
Response: We find this to be a fascinating point, prompting us to review our AYA STS experience for patients with similar diagnoses/expectations that were otherwise not eligible for the study. We identified a 29 yer old female with widely metastatic ASPS (described previously: PMID28544595) who was privately insured and succumbed to disease more quickly despite access to many of the same therapies delivered to the patient with ASPS described in our cohort. We have diagnosed and treated a privately insured 25 year old male with widely metastatic EHE who was treated with pazopanib for one year and is now in the first year of off-therapy monitoring. He was not yet eligible for the present study. Another privately insured EHE patient (39 year old female) is being monitored after resection of her primary mass and has suspicion of metastatic lung nodules, but metastasis has not been proven. Another metastatic ES patient (male in his 30s) with private insurance was treated on a clinical trial after disease progression following upfront therapy at his home institution but did not live more than 2 years from his original diagnosis.
The possibility of selection bias certainly still exists. Conceivably, there are publicly insured (and privately insured) patients with other metastatic STS diagnoses that could have lived > 2 years from diagnosis, had there been no limitations in therapeutic options, but it would require a great deal of speculation to identify any such patients. On the other hand, we would point out that Patient 3, who had the most consecutive lines of therapy (8 lines) and one of the longest treatment durations (6 years) was the patient without private insurance.
Round 2
Reviewer 1 Report
Comments and Suggestions for Authors
The revised manuscript demonstrates significant improvements in the depth of qualitative analysis, discussion, and conclusion sections that I previously identified as areas requiring enhancement. Although increasing the sample size remains challenging, this limitation has been adequately addressed by indicating it as a direction for future research. Thus, my initial concerns have been successfully resolved.